# Neuregulin 2 Is a Candidate Gene for Autism Spectrum Disorder

**DOI:** 10.3390/ijms25105547

**Published:** 2024-05-19

**Authors:** Wei-Hsien Chien, Chia-Hsiang Chen, Min-Chih Cheng, Yu-Yu Wu, Susan Shur-Fen Gau

**Affiliations:** 1Department of Occupational Therapy, College of Medicine, Fu Jen Catholic University, New Taipei City 242062, Taiwan; 2Department of Psychiatry, Linkou Chang Gung Memorial Hospital, Taoyuan 333, Taiwan; cchen3801@gmail.com (C.-H.C.); ruthyuyuwu@gmail.com (Y.-Y.W.); 3Department of Psychiatry, Yuli Branch, Taipei Veterans General Hospital, Hualien 981, Taiwan; cmc@mail.vhyl.gov.tw; 4Department of Psychiatry, National Taiwan University Hospital, Taipei 10002, Taiwan; 5Graduate Institute of Brain and Mind Sciences and Graduate of Clinical Medicine, National Taiwan University, Taipei 10002, Taiwan

**Keywords:** autism spectrum disorders, neuregulin 2 (NRG2), insertion/deletion polymorphism (indel)

## Abstract

Autism spectrum disorder (ASD) is a complex neurodevelopmental disorder with heterogeneous and complex genetic underpinnings. Our previous microarray gene expression profiling identified significantly different neuregulin-2 gene (NRG2) expression between ASD patients and controls. Thus, we aimed to clarify whether NRG2 is a candidate gene associated with ASD. The study consisted of two stages. First, we used real-time quantitative PCR in 20 ASDs and 20 controls to confirm the microarray gene expression profiling results. The average NRG2 gene expression level in patients with ASD (3.23 ± 2.80) was significantly lower than that in the controls (9.27 ± 4.78, *p* < 0.001). Next, we conducted resequencing of all the exons of NRG2 in a sample of 349 individuals with ASD, aiming to identify variants of the NRG2 associated with ASD. We identified three variants, including two single nucleotide variants (SNVs), IVS3 + 13A > G (rs889022) and IVS10 + 32T > A (rs182642591), and one small deletion at exon 11 of NRG2 (delGCCCGG, rs933769137). Using data from the Taiwan Biobank as the controls, we found no significant differences in allele frequencies of rs889022 and rs182642591 between two groups. However, there is a significant difference in the genotype and allele frequency distribution of rs933769137 between ASDs and controls (*p* < 0.0001). The small deletion is located in the EGF-like domain at the C-terminal of the NRG2 precursor protein. Our findings suggest that NRG2 might be a susceptibility gene for ASD.

## 1. Introduction

According to the Diagnostic and Statistical Manual of Mental Disorders 5 (DSM-5), autism spectrum disorder (ASD) is defined as a childhood-onset neurodevelopmental disorder with two essential features: persistent deficits in social communication and interactions, and restricted, repetitive patterns of behavior, interests, or activities [1]. Although the prevalence of ASD in Taiwan was estimated to be 1% in 2017 [2], the figure has markedly increased in the past decade. Recent data showed that around one in 36 people (2.8%) aged 8 years have been diagnosed with ASD come from 11 communities in the Autism and Developmental Disabilities Monitoring Network in the United States published in 2023 CDC’s Morbidity and Mortality Weekly Report [3], and that males are more frequently affected than females, with a male-to-female ratio of approximately 4.3:1 [4,5]. The heritability of ASD is estimated to be >90%, suggesting a strong genetic component [6]. Recently, large-scale gene discovery efforts have shown that ASD is not a simple Mendelian disorder [7,8,9]. Conversely, the genetic underpinnings of ASD are heterogeneous and complex, involving multiple genes as well as gene–gene and gene–environment interactions [10]. The genetic risk factors for ASD identified so far range from common variants conferring a small clinical effect to rare mutations with a high clinical outcome [11]. Recently, the application of large-scale exome sequencing has led to the identification of hundreds of novel ASD-associated genes enriched in common genetic signaling pathways such as synaptic development, plasticity, and signaling [12]. The high heterogeneity of ASD may account for the varied clinical presentations of patients with ASD. Despite these advances, there are likely more ASD-associated genes to be discovered.

Microarray-based gene expression profiling allows simultaneous measurement of hundreds to thousands of gene transcripts, useful for large-scale gene discovery [13]. Comparative gene expression profiling analysis of lymphoblastoid cell lines (LCL) has identified shared pathways among different forms of autism [14]. Altered pathways in neural development and steroid biogenesis were detected in a total gene expression profiling analysis of LCL in autistic patients and their unaffected siblings [15]. Previously, our group conducted comparative gene expression profiling of LCL in ASD patients and control subjects to identify differentially expressed genes associated with ASD [16]. A total of 187 differentially expressed probe sets, including 131 transcripts (50 up-regulated and 81 down-regulated), were detected between cases and controls. We verified one of the differentially expressed genes, FOXP1, in a sample of ASD patients and control subjects using real-time quantitative polymerase chain reaction (RT-qPCR). The average expression level of FOXP1 in ASD was significantly higher than that of the controls [16].

Neuregulin 2 (NRG2) emerged as a notable differentially expressed gene during the comparative total gene expression profiling analysis in our previous study [16]. NRG2 belongs to the NRG gene family, which includes NRG1–NRG4. NRGs serve as cell–cell ligands for receptor tyrosine kinases belonging to the ErbB family, which comprises four homologous type I receptor tyrosine kinases known as EGFR (epidermal growth factor receptor), ErbB1, ErbB2, ErbB3, and ErbB4 [17,18]. The neuregulin–ErbB signaling network is involved in several processes in both the developing and adult brain [19]. NRGs specifically play a role in synaptic plasticity, promoting neuronal migration and differentiation, regulating the expression of neurotransmitter receptors, and influencing glial proliferation, survival, and differentiation [19,20]. Through previous genetic and functional analytic studies, several variants of NRG1 and its neuronal receptor ErbB4 were found to be associated with schizophrenia and its endophenotypes [21,22,23]. Moreover, studies in human-induced pluripotent stem cells from affected subjects demonstrated that the NRG–ErbB pathway is related to psychiatric disorders [24,25,26]. In a recent study, Yanan Deng reported that inhibition of the Notch pathway using DAPT alleviated autism-like behaviors and reduced NRG1 and phosphorylated ErbB4 levels, suggesting the involvement of the Notch1/Hes1 pathway in regulating the NRG1/ErbB4 pathway in autism [27].

NRG2, encoding a novel member of the NRG family, induces the growth and differentiation of epithelial, neuronal, glial, and other cell types through interaction with ErbB receptors while sharing a similar genomic structure with NRG1 [28]. Like NRG1, NRG2 can bind to ErbB4 and ErbB3 but has no active kinase domain [21]. NRG2 is expressed in the embryonic heart, lung, and bladder, and the developing nervous system [29,30] but is confined to the cerebellum (granule cells and Purkinje cells), dentate gyrus (granule cells), and olfactory bulb (granule cells) in the adult brain [29,30,31]. Recently, Vullhorst et al. reported that NRG2, but not NRG1, is a major functional ErbB4 ligand in the postnatal brain controlling the N-methyl-D-aspartate (NMDA) receptor function in cortical interneurons, and is associated with cognitive deficits in psychiatric disorders [32]. Recently, Yan et al. found higher dopamine levels in the dorsal striatum but lower levels in the medial prefrontal cortex (mPFC) in Nrg2 knockout mice (KO). The pattern of dopamine expression was similar to schizophrenia, and animals showed behavioral abnormalities relevant to psychiatric disorders, such as impaired social behavior and cognitive function [33]. Further, a genome-wide association analysis of disturbances in the electroencephalography early gamma-frequency band between schizophrenia and control subjects found significant differences in several markers of the NRG2 and KALRN genes involved in neuronal development and the NRG–ErbB signaling pathway [34]. Furthermore, several studies have shown that NRG1, NRG2, or genes of the NRG–ErbB network contribute to the etiology of psychiatric disorders [35,36,37]. Nevertheless, the function of NRG2 is not well known, and its association with complex neurodevelopmental disorders like ASD remains unclear.

NRG2 (Gene ID 9542) is located at chromosome 5q31.2, an ASD candidate gene region, as shown by genetic linkage studies [38,39]. In addition, several case reports indicated that cases with ASD have de novo translocations of chromosome 5q and chromosomes 1q, 4q, and 18q [40,41,42]. Furthermore, Phillippi et al. reported a strong and consistent association between two Single-Nucleotide Polymorphisms (SNPs) within the paired-like homeodomain transcription factor 1 (PITX1) on chromosome 5q31 and autism [43].

Based on the above findings, we suggest that NRG2 might be a susceptibility gene for ASD. To test this hypothesis, we first used RT-qPCR to assess the NRG2 expression level in a sample of 20 patients with ASD and 20 controls to verify the microarray gene expression analysis results. Next, we conducted Sanger sequencing of all the exons of NRG2 in an independent sample of ASD patients, aiming to identify variants that may be associated with ASD.

## 2. Results

### 2.1. Differential Expression of NRG2 in ASD Patients and Controls

We first compared the mRNA levels of NRG2 in the LCL in 20 male subjects with ASD and 20 healthy male controls using real-time quantitative PCR (RT-qPCR). As shown in Figure 1, the average mRNA level of NRG2 in the ASD subjects (3.23 ± 2.80) was significantly lower than that in the control subjects (9.27 ± 4.78, *p* < 0.001).

### 2.2. Detection of Three Variants of NRG2

We further conducted Sanger sequencing of all the exons and their junction sequences of NRG2 in 349 patients with ASD to seek molecular variants that may explain the lower NRG2 expression in ASD. We identified three variants, including IVS3 + 13A > G, IVS10 + 32T > A, and a small deletion GCCCGG at the last exon of NRG2. The locations of these three variants and their chromatographs are illustrated in Figure 2. 

### 2.3. Association Analysis of Three Variants with ASD

All these three variants have been reported in dbSNP, and have been designated rs889022 (IVS3 + 13A > G), rs182642591 (IVS10 + 32T > A), and rs933769137 (deletion of GCCCGG), respectively. To find out whether these three variants were associated with ASD, we compared the allelic frequencies of these three variants detected from ASD subjects with those reported in the Taiwan Biobank (https://taiwanview.twbiobank.org.tw/, accessed on 16 May 2024) as the controls. We found no significant differences in the allelic and genotype distributions of rs889022 and rs182642591 between two groups. However, we found significant differences in the allelic frequencies between ASD patients and the controls (*p* < 0.0001). A total of 31 patients out of 264 patients (11.7%) were heterozygotes carrying the small deletion, while none of 1492 controls were heterozygous carriers. The significant differences in the allele frequencies remained after correction for multiple testing. The data are listed in Table 1.

### 2.4. Functional Prediction of the Small Deletion rs933769137

The small deletion, GCCCGG, occurred at the coding region of the last exon of NRG2, which led to in-frame deletion of two amino acids (Proline and Glycine) at amino acid sequences 674 and 675 (Reference Sequence: NP_053585.1). The small deletion was predicted to have a damaging effect by SIFT, while PROVEAN predicted that this small deletion was a benign mutation. (Table 2).

### 2.5. Power Analysis

A post-hoc power analysis showed that with a total sample size of 299 participants, excluding about 50 cases without DNA, we had a power of 0.32 to detect a small effect (0.1) and 0.998 to see a medium effect (0.3) of the genotype distributions of IVS3 + 13, IVS10 + 32, and the small deletion GCCCGG variant in exon 11 at an alpha level of 0.05. To analyze the allele frequencies of these three variants, we had a power of 0.32 to detect a small effect (0.1) and 0.998 to see a medium effect (0.3) at an alpha level of 0.05.

## 3. Discussion

In our study, we found a significantly lower average NRG2 expression in ASD patients (3.23 ± 2.80) than that in controls (9.27 ± 4.78, *p* < 0.001; Figure 1). Previously, several linkage analysis and association studies pointed to a genome region encompassing the NRG2 locus to be associated with psychiatric disorders [44,45,46]. In addition, many studies have pointed out that 5q containing the NRG2 gene may also be a candidate region for ASD [38,39,40,41,42,43]. Recently, Yan L. et al. reported that NRG2-KO mice presented a pattern of dopamine that was similar to schizophrenia and also showed various behavioral abnormalities relevant to psychiatric disorders, such as impaired social behavior and cognitive function in several behavioral tests [33]. Our findings align with prior research and indicate that NRG2 could be a susceptibility gene for ASD, and the reduced NRG2 expression or its loss of function may be associated with the development of ASD.

Furthermore, the small deletion GCCCGG led to in-frame deletion of proline and glycine at amino acid sequences 674 and 675 of NRG2 (Reference Sequence: NP_053585.1), which are part of the EGF-like domain region at the C-terminal of the NRG2 precursor protein as predicted in silico (UniProt). Previous studies have shown that the EGF-like domain regulates the biological activity of NRG2 [19,29], as it is the region that binds to receptors and can regulate processes such as cell proliferation, differentiation, and survival. The function of the protein C-terminal domain includes regulation of protein stability or function or interactions with other proteins. Recently, Czarnek and Bereta reported the proteolytic degradation mechanism of NRG2 protein; the C-terminal part contains a site for γ-secretase. They are cleaved by ADAM10 or BACE2 for degradation. The interaction domains in the C-terminal region of NRG2 lie at 712–713 amino acids. This domain interacts with other neurotrophic factors; these interactions might help regulate physiological and pathological processes such as neural development and neurodegeneration [47].

To gain insights into the evolutionary conservation of the neighboring amino acids surrounding this small deletion variant, we conducted sequence alignment analyses across multiple mammalian species available in the NCBI database (https://blast.ncbi.nlm.nih.gov/Blast.cgi, accessed on 16 May 2024). Hence, we observed the presence of this variant in certain mammalian species as well. This observation suggests that the small deletion of nucleotide sequences may not be conserved across different species. Consequently, it is plausible to suggest that this deletion might not directly impact the function of NRG2. However, it is worth noting that the del (GCCCGG) variant was significantly more common among our ASD cases compared to the control population. Therefore, to comprehensively understand the involvement of the small deletion in ASD, it is essential to conduct additional functional analysis that provides valuable insights into the underlying physiological and pathological mechanisms.

This study still has some limitations. First, we identified two intronic SNVs (rs889022 and rs182642591) in nearly 300 ASD Taiwanese patients, but we did not find a significant association with ASD in our autistic samples. Population-based case-control association studies of complex conditions, like ASD with clinical and genetic heterogeneity, need a large sample size to identify risk genes with small effects. In this study, we recruited 349 ASD patients and 1492 genetic data of the Taiwan Biobank as controls into this study. Due to the challenge of obtaining blood samples from children with normal development, we opted to utilize genetic data obtained from the Taiwan Biobank as our control group. Moreover, our post-hoc power analysis revealed a power of 0.32 for detecting a small effect size (0.1) and a power of 0.998 for detecting a medium effect size (0.3) in terms of the genotype distributions and allele frequencies of the three identified variants. Therefore, further studies with larger sample sizes or meta-analysis are needed to address the association of the NRG2 gene and ASD. Second, despite the significantly lower NRG2 expression level in ASD patients than in 20 controls, the regulatory mechanism underlying its potential influence on ASD remains unknown. We could not address causality in this study. Future studies should focus on the regulatory mechanism governing the effects of lower NRG2 expression on ASD. Finally, we conducted in silico analysis to explore the potential deleterious effects of this small deletion variant. The variant was predicted to have damaging effect by SIFT, while PROVEAN predicted that this small deletion is a benign mutation. However, the function of the small deletion of the EGF-like domain at the C-terminal region of NRG2 and how it modulates the protein-protein interaction in the NRG-ErbB pathway still remains unknown. Therefore, further functional analysis of the small deletion is necessary to gain insights into the physiological and pathological mechanisms underlying its involvement in ASD.

## 4. Materials and Methods

### 4.1. Subjects

The study was conducted in two stages. First, an RT-qPCR analysis was performed in ASD cases and controls. NRG2 transcription levels were compared between 20 patients with ASD and 20 healthy controls. The 20 autistic patients were all males aged 9.1 ± 3.2 years (4–18), and the 20 male controls were 37.15 ± 16.90 years (18–67) recruited among the cohort of ASD patients receiving regular medical checkups in a local medical center. Second, 349 additional ASD patients were recruited for exonic resequencing analysis, including 308 male patients (15.26 ± 5.23 years old) and 41 female patients (14.88 ± 4.90 years old). All the patients with ASD were recruited from the Children Mental Health Center, Department of Psychiatry, Taiwan University Hospital (NTUH), Taipei, Taiwan, and the Department of Child Psychiatry, Chang Gung Memory Hospital (CGMH), Kwei-Shan, Taiwan. The clinical diagnosis of ASD made by qualified child psychiatrists was confirmed by interviewing the caregivers using the Chinese version of the Autism Diagnostic Interview-Revised [48,49] and patients using the Chinese version of the Autism Diagnostic Observation Scale [50]. Patients with known chromosomal abnormalities or associated medical conditions were excluded from the study. Finally, the super-controls from the Han Chinese Cell and Genome Bank established by the Institute of Biomedical Sciences, Academia Sinica in Taiwan [51] were used for case-control association analysis. The study protocol was approved by the Research Ethics Committee of NTUH, CGMH, and Fu Jen Catholic University; written informed consent was obtained from the participants and their parents after the procedures were fully explained and before study implementation, including blood sample collection for subsequent laboratory investigations.

### 4.2. Preparation of the Lymphoblastoid Cell Line and cDNA

Immortalized LCLs were established from each subject by transforming lymphocytes with Epstein–Barr virus following the procedures described elsewhere [52]. The cDNA of 20 ASD participants and 20 controls was prepared using SuperScriptTM II RNase H- Reverse Transcriptase (Invitrogen Life Technologies, Carlsbad, CA, USA). The details have been described in our previous report [52].

### 4.3. Real-Time Quantitative PCR

RT-qPCR was performed using the SYBR Green method and implemented in the StepOne Plus Real-Time PCR System according to the manufacturer’s protocol (Applied Biosystems, Forster City, CA, USA). Detailed procedures can be found in our previous report [51]. The relative standard curve method was used to quantify mRNA expression (User Bulletin #2 ABI PRISM 7700 sequence detection system). This method used serial dilutions of known RNA amounts from a reference sample (pooled from 40 LCLs of male controls) to generate an external standard curve. The relative amount was calculated using linear regression analysis from their respective common angles for each unknown piece. The mRNA expression level of NRG2 was normalized by the 18S rRNA. The 18S rRNA reference gene was measured using pre-developed TaqMan assay reagents 18S rRNA MGB according to the manufacturer’s protocol (Applied Biosystems, Foster City, CA, USA). All experiments were performed in two independent sets to ensure reproducibility. Primer sequences for PCR amplification were designed using online Primer3 software (version 4.1.0) (http://bioinfo.ut.ee/primer3/, accessed on 16 May 2024) and listed in Table 3.

### 4.4. DNA Purification

Genomic DNA was prepared from peripheral blood using the Puregene DNA purification system (Gentra Systems Inc., Minneapolis, MI, USA) according to the manufacturer’s protocol or from saliva using the Oragene DNA self-collection kit (DNA Genotek, Ottawa, ON, Canada) following the manufacturer’s instructions.

### 4.5. PCR Amplification

The genomic sequences of the human neuregulin 2 gene were available from the NCBI (https://www.ncbi.nlm.nih.gov/). NRG2 (Gene ID: 9542) comprises 11 exons with a genomic size of 196,519 bp (RefSeq: NM_013982.3). Optimal PCR primer sequences were generated to amplify each NRG2 exon using Primer3 (version 4.1.0) (http://bioinfo.ut.ee/primer3/, accessed on 16 May 2024) (Table 4). In the standard reaction, genomic DNA (75 ng) was amplified in a reaction volume of 15 μL containing 0.75 μM each of sense and antisense primers, 0.15 mM of dNTP, 50 mM of KCl, 1.5 mM of MgCl_2_, 0.1% vol/vol of Triton X-100, 10 mM of Tris–HCl (pH 9.0), and 1 U Taq polymerase. PCR conditions consisted of an initial denaturation at 95 °C for 5 min, followed by 30 cycles of 95 °C for 1 min, the optimal annealing temperature of detected amplicons for 1 min, and 72 °C for 1 min. PCR was performed with a PTC-200 DNA engine (MJ Research, Watertown, MA, USA). Each amplicon was subjected to direct sequencing using the ABI Prism™ BigDye™ Terminator Cycle Sequencing Ready Reaction Kit Version 3.1 and ABI auto sequencer 3730 (Perkin-Elmer Applied Biosystem, Foster City, CA, USA), according to the manufacturer’s protocol.

### 4.6. Direct PCR Autosequencing

After PCR amplification, aliquots of PCR products were processed using the PCR Pre-sequencing Kit (USB Corp. Cleveland, OH, USA) to remove residual primers and dNTPs according to the manufacturer’s instructions. The purified PCR products were subjected to direct sequencing using the ABI Prism™ BigDye™ Terminator Cycle Sequencing Ready Reaction Kit Version 3.1 and the ABI autosequencer 3730 (Perkin-Elmer Applied Biosystem, Foster City, CA, USA) according to the manufacturer’s protocol.

### 4.7. Statistical Analysis

Differences in allele and genotype frequencies of NRG2 between patients and controls were assessed by the chi-square test or Fisher’s exact test where appropriate. Assessment of pair-wise linkage disequilibrium (D′) of NRG2 SNPs was implemented using SHEsis computer program [53]. A *p*-value of <0.05 was considered statistically significant. Moreover, post-hoc power analysis was performed using the computer program G*Power [54].

### 4.8. In Silico Analysis

Prediction of potential deleterious effects of NRG2 SNPs was performed using the software tools SIFT (UPDATE on 25 April 2024, https://sift.bii.a-star.edu.sg/, accessed on 16 May 2024), UniProt (Release 2024_02, https://www.uniprot.org/, accessed on 16 May 2024), and PROVEAN v1.1.3 (http://provean.jcvi.org/index.php, accessed on 16 May 2024). PROVEAN web server functions are currently using PROVEAN v1.1.3.

## 5. Conclusions

We confirmed that the average gene expression level of NRG2 in LCL is significantly lower in ASD patients compared to control subjects. Furthermore, this study identified three variants in NRG2, including two noncoding SNVs and one small deletion in the coding region. There were no significant differences in the genotype or allele frequency distributions of the first two noncoding dbSNVs between ASD patients and controls. In contrast, the genotype and allele frequency distribution of the indel variant showed significant differences between ASD cases and controls. Together, this study suggests that NRG2 may be a candidate gene associated with ASD in Taiwan. This finding may have valuable implications for comprehending the etiology, diagnosis, and treatment of patients with ASD in the future. Moreover, our research offers an effective approach to finding the genes associated with complex diseases such as schizophrenia and ASD.

## Figures and Tables

**Figure 1 ijms-25-05547-f001:**
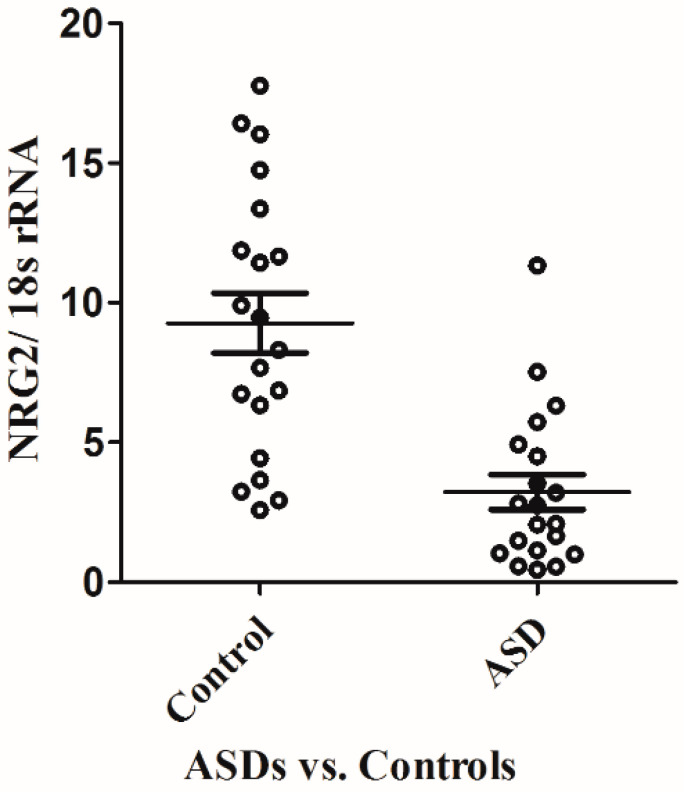
Scatter plot of the NRG2 mRNA level as normalized by 18S rRNA in 20 patients with Autism Spectrum Disorder (ASD) and 20 control subjects. The horizontal line indicates the mean of NRG2 transcript. The average expression level of the NRG2 gene transcript in patients with ASD (3.23 ± 2.80) was significantly lower than that of control subjects (9.27 ± 4.78), *p* < 0.001.

**Figure 2 ijms-25-05547-f002:**
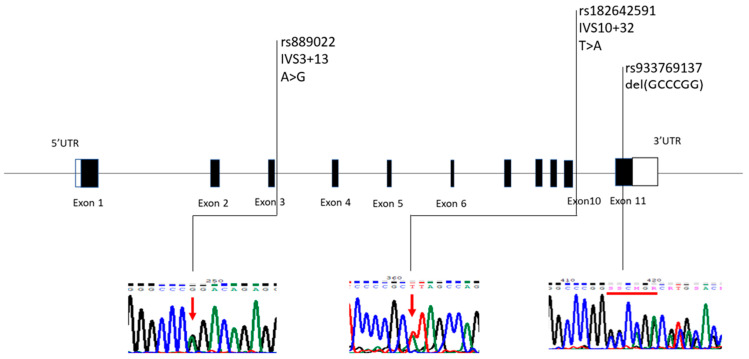
Schematic genomic structure of NRG2, and locations of the variants identified in this study. NRG2 (Gene ID: 9542) comprised 11 exons; the genomic size is 196,519 bp on chromosome 5q31.2. (RefSeq: NM_013982.3). The three variants, IVS3 + 13 (rs889022), IVS10 + 32 (rs182642591), and delGCCCGG (rs933769137) we identified were also dbSNPs. The dbSNPs of NRG2 amplicons were available from NCBI (https://www.ncbi.nlm.nih.gov/). The locations of these variants are listed in the figure. The results of the sequencing of these three variants are shown below in Figure 2 (red arrows and red line).

**Table 1 ijms-25-05547-t001:** Genotype and allele frequencies of variants of the NRG2 gene identified in ASD patients vs. controls (from the Taiwan biobank).

SNP		Location	Groups	n	Genotype			*p* Value	Allele		*p* Value
rs889022		139880843			A/A	A/G	G/G		A	G	
			ASD Control	231 1492	197 (85.3%) 1191 (79.8%)	30 (13.0%) 283 (19.0%)	4 (1.7%) 18 (1.2%)	0.08	424 (91.8%) 2670 (89.3%)	38 (8.2%) 320 (10.7%)	0.10
rs182642591		139851572			T/T	T/A	A/A		T	A	
			ASD Control	299 1492	298 (99.7%) 1470 (98.5%)	1 (0.3%) 22 (1.5%)	0 0	0.11	597 (99.8%) 2968 (99.3%)	1 (0.2%) 22 (0.7%)	0.11
rs933769137		139848470–139848475			GCCCGG/GCCCGG	GCCCGG/del(GCCCGG)	del(GCCCGG)/del(GCCCGG_)_		GCCCGG	del(GCCCGG)	
			ASD	264	233 (88.3%)	31 (11.7%)	0	<0.0001 **	497 (94.1%)	31 (5.9%)	<0.0001 **
			Control	1492	1492 (100.0%)	0 (0.0%)	0		2984 (100.0%)	0 (0.0%)	

The rs933769137 was also named rs1323957797, rs1284917723, and rs1226370164 located at chromosome 5: 139,848,470 (Ref. Version: GRCh38). The rs1323957797, rs1284917723, and rs1226370164 have merged into rs933769137 (Release Version: 20230706150541). Control: 1492 genetic data samples from the Taiwan Biobank as controls (https://taiwanview.twbiobank.org.tw/). *p* value ** < 0.01.

**Table 2 ijms-25-05547-t002:** Variants of the NRG2 gene identified in ASD patients and controls and their functional predictions.

	Location	a.a Position	Variants	Consequence Type	In Silico Analysis			Amino Acid Change
					SIFT	PROVEAN (Cutoff = −2.5)	UniProt	
rs933769137	Ch5: 139848470–139848475	674–675	del(GCCCGG)	In-frame deletion	damaging	neutral	Likely Benign	GPGPGpgADMQR-GPGPGADMQR

Lowercase letters represent the changed amino acids. Reference amino acids sequence: NP_053585.1 (NCBI Reference Sequence). PROVEAN (Protein Variation Effect Analyzer) is a software tool which predicts whether an amino acid substitution or indel has an impact on the biological function of a protein. PROVEAN web server functions are currently using PROVEAN v1.1.3. (http://provean.jcvi.org/index.php, accessed on 16 May 2024).

**Table 3 ijms-25-05547-t003:** Sequences of primers used in the real-time quantitative PCR (RT-qPCR) experiments in NRG2.

Gene	Forward Primer	Backward Primer	Ta
NRG2	CCACAGACCATGTCATCAGG	CCGACTGGGAGTCAGAAGTC	60

Ta, annealing temperature. NRG2 [Gene ID: 9542; GenBank: NM_013982.3].

**Table 4 ijms-25-05547-t004:** Primer sequences, optimal annealing temperature (Ta), and size of PCR products of the NRG2 gene.

Amplicon	Forward	Reverse	Ta (°C)	Size (bp)
Exon 1.1	TTACGCTGTTTCCGGTTTTC	TGGTCCTGCACTGACTTGAG	60	458
Exon 1.2	GCTTCTCCATGCTGCTCTTC	ttcttctctccaaccccaac	58	586
Exon 2	acagtggcccttactctcca	ctggttccatgggtgagtct	63	372
Exon 3	agggaatctccttcccatct	gttgagtgcgagatggatca	63	356
Exon 4	gagatgattcctggggccta	acttctgacccagcatctcc	60	250
Exon 5	ccaagtgcctgacttggttt	tgcacccagaagctttctaa	63	244
Exon 6	ggagtcctgaccaacgtttc	cattcagcacacatggcatc	60	208
Exon 7	aaggggtctctgcaccacta	acattcttggaggcccatc	67	238
Exon 8–9	gaagttcatcgttggcgagt	ggtgtgctgtgattcctgtg	67	786
Exon 10	gagtggagaagggcattgag	atggagatgaggctctttgg	67	468
Exon 11	gttatgcccgcgtaacagat	CCCAGATGAGCATACAGCAA	60	1343

Ta, annealing temperature. NRG2 [Gene ID: 9542; GenBank: NM_013982.3].

## Data Availability

The raw data are available upon request of the corresponding author. All SNPs information on controls were from the Taiwan Biobank (https://taiwanview.twbiobank.org.tw/). Primers are designed using online Primer3 software (http://bioinfo.ut.ee/primer3/). Prediction of potential deleterious effects of SNPs performed using the software tools were PROVEAN (http://provean.jcvi.org/index.php) and SIFT (http://sift.jcvi.org/) and UniProt (https://www.uniprot.org/).

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
