# Peer review of "Neuregulin 2 Is a Candidate Gene for Autism Spectrum Disorder"

_ijms, 2024, doi:10.3390/ijms25105547_

Round 1
Reviewer 1 Report
Comments and Suggestions for Authors
The authors follow-up on previous expression studies revealing an association between NRG2 and autism, again showing an (impressive) association. This was followed with sequencing studies showing that an indel in that gene is strongly associated with autism. Together, there is a strong case that NRG2 is a risk factor for autism.
I have no major concerns with the manuscript, only some minor ones:
1. The GCCCGGC indel appears to be 7 nucleotides long, which would create a frameshift (indeed, that is stated in Table 4). However, the heterozygous condition is listed as C/GCCCGGC, which appears to be 6 nucleotides long, which would create an in-frame insertion of 2 amino acids. There is a big difference! In any event, and especially in case of the frameshift, the usual in silico prediction algorithms would not be useful. What amino acids are added (if in frame)? What percentage of the coding sequence is missing (if frameshift)? What is the evolutionary conservation of amino acids in the vicinity of the indel (if in frame)? Figure 2 suggests that the indel is at the end of the last exon (10), so a frameshift may not remove many amino acids. Are those terminated amino acids highly conserved?
2. One limitation (as well as a strength) was that the study was only conducted in Taiwan. What is the prevalence of the GCCCGGC indel in general and/or in other major populations?
Author Response
Reviewer 1
Comment: 1. The GCCCGGC indel appears to be 7 nucleotides long, which would create a frameshift (indeed, that is stated in Table 4). However, the heterozygous condition is listed as C/GCCCGGC, which appears to be 6 nucleotides long, which would create an in-frame insertion of 2 amino acids. There is a big difference! In any event, and especially in case of the frameshift, the usual in silico prediction algorithms would not be useful. What amino acids are added (if in frame)? What percentage of the coding sequence is missing (if frameshift)? What is the evolutionary conservation of amino acids in the vicinity of the indel (if in frame)? Figure 2 suggests that the indel is at the end of the last exon (10), so a frameshift may not remove many amino acids. Are those terminated amino acids highly conserved?
Reply: After careful checking, the indel variant we found is actually a deletion of 6 nucleotides GCCCGG, thanks for your correction. The small deletion was designated (GCCCGG) in our revised manuscript. The deletion resulted in an in-frame deletion of two amino acids Proline and Glycine at the positions 674 and 675 of the NRG2 amino acid sequences (Reference Sequence: NP_053585.1). Also, we have updated the reference genomic sequences of NRG2 to NM_013982.3 (GRCh38). Hence, the scheme of genomic structure of NRG2 and the positions of all the variants found in this study were revised accordingly. There are different alleles of indel variants at the same location in dbSNP with our deletion, which was given reference number rs933769137 the dbSNP (https://www.ncbi.nlm.nih.gov/snp/). Hence, we also use this reference number in our revised manuscript.
To assess the evolutionary conservation of amino acids in the vicinity of the identified indel variant, we compared various species of mammals in the NCBI database, presented the following information: (https://blast.ncbi.nlm.nih.gov/Blast.cgi)
In our study, we identified a small deletion variant involving two amino acids, Proline and Glycine, at positions 674 and 675 within the NRG2 amino acid sequences (Reference Sequence: NP_053585.1). This variant was also detected in certain mammals. This observation suggests that the small deletion of nucleotide sequences may not be conserved across different species. (page6_lines 198-206 in our revised manuscript).
- One limitation (as well as a strength) was that the study was only conducted in Taiwan. What is the prevalence of the GCCCGGC indel in general and/or in other major populations?
Reply: Source from NCBI SNP database:
According to the dbSNP (rs933769137 RefSNP Report - dbSNP - NCBI (nih.gov), the allele frequence of indel variant at this position is 0 in a total sample size of 14044 in the ALFA project. But in 1000Genomes-30X project, the allele frequency of CCCGGG deletion was 0.0025 in a smaple size of 1202 of South Asian. In Taiwan BioBank, in a sample of 1492 with Illumina whole genome sequencing, an insertion of GCCGGGC was reported with an allele frequency of 0.001 (Taiwan BioBank (twbiobank.org.tw). Hence, the Del (GCCCGG) found in our study is quite unique and needs further replication studies.

Reviewer 2 Report
Comments and Suggestions for Authors
Dear Authors
After reviewing The article titled "Neuregulin 2 is a Candidate Gene for Autism Spectrum Disorder" presents an interesting study on the potential genetic underpinnings of Autism Spectrum Disorder (ASD) through the investigation of Neuregulin 2 (NRG2). While the study adds to the growing body of knowledge surrounding the genetic factors involved in ASD, several critical parts require attention for improvement:
- Study Design and Sample Size:
- The sample size of only 20 ASD patients and 20 controls for real-time quantitative PCR analysis is relatively small, limiting the generalizability of the findings.
- A larger cohort is recommended for future studies to validate the results more robustly.
- Control Group Selection:
- There is a notable mismatch in age and gender distribution between the ASD group and the control group, potentially introducing confounding variables.
- Future studies should ensure that control groups are appropriately matched to the experimental group in terms of age, sex, and other relevant demographics.
- Statistical Analysis:
- The article does not provide detailed information on the adjustments made for multiple comparisons, which is crucial given the multiple genetic analyses performed.
- Adequate statistical corrections are necessary to avoid the risk of false-positive results.
- Functional Analysis of Identified Variants:
- Although several genetic variants in NRG2 were identified, the paper lacks comprehensive functional analyses to elucidate their roles in ASD.
- It is essential to conduct functional studies to establish a causal link between these genetic variants and the pathology of ASD.
- Interpretation of Results:
- The discussion tends to overstate the implications of the genetic findings, suggesting NRG2 as a susceptibility gene for ASD without sufficient supporting evidence.
- More detailed genetic and functional analyses are required to substantiate such claims.
In conclusion, while the study conducted is a valuable step towards understanding the genetic landscape of ASD, the concerns outlined above regarding the study's design, methodology, and interpretation of results highlight the need for more rigorous approaches.
Regards
Author Response
Comment: 1. Study Design and Sample Size:
The sample size of only 20 ASD patients and 20 controls for real-time quantitative PCR analysis is relatively small, limiting the generalizability of the findings.
A larger cohort is recommended for future studies to validate the results more robustly.
Reply:
The RT-qPCR was aimed to confirm our previous differential expression study between ASD and controls using expression microarray. Despite the small sample sizes, we were able to confirm the differential expression of NRG2 in LCL between 20 ASD and 20 controls using RT-qPCR, we think we have good reason to invest our resources to look for variants of NRG2 in ASD that may explain the reduced NRG2 expression in LCL. Certainly, replication studies with larger sample size would provide better power to verify whether the differential expression is true or false positive finding.
2.Control Group Selection:
There is a notable mismatch in age and gender distribution between the ASD group and the control group, potentially introducing confounding variables.
Future studies should ensure that control groups are appropriately matched to the experimental group in terms of age, sex, and other relevant demographics.
Reply:
Thank for the comment. We know we had better use matched controls in this study regarding sex, age and other confounders. But in reality, it is difficult in obtain blood sample from children with normal development. Hence, we use the genetic data from Taiwan Biobank as the controls. We have stated this as a limitation in our discussion.
3.Statistical Analysis:
The article does not provide detailed information on the adjustments made for multiple comparisons, which is crucial given the multiple genetic analyses performed.
Adequate statistical corrections are necessary to avoid the risk of false-positive results.
Reply: Thank you for your correction. We have provided the corrections in the revised manuscript (Page5_lines140-152). The rs889022 SNP was found in 14.7% (34/231) of ASD patients and 20.2% (302/1492) of control subjects (p = 0.08), and the rs182642591 was found in 0.3% (1/ 299) of ASD patients and 1.5% (22/1492) of controls (p = 0.11). There were no significant differences in the allelic and genotype distributions of rs889022 and rs182642591 between two groups. The genotype and allele frequency distribution of the indel variant was significantly differed between ASD patients and controls. A total of 31 patients out of 264 patients (11.7%) were heterozygotes carrying the small deletion, while none out of 1492 controls were heterozygous carriers. The significant differences in the allele frequencies remained after correction for multiple testing. The data are listed in Table 3.
- Functional Analysis of Identified Variants:
Although several genetic variants in NRG2 were identified, the paper lacks comprehensive functional analyses to elucidate their roles in ASD.
It is essential to conduct functional studies to establish a causal link between these genetic variants and the pathology of ASD.
Reply: We know it is essential to conduct functional assay of the del (GCCCGG) found in this study, but due to the restraint of our expertise and resources, we can only conduct in silico analysis to explore the potential deleterious effects of NRG2 variants. The results are presented in Table 4 of this revised manuscript. (page5_lines 154-158; page11_table4). Also, we stated this point as one of the limitations in the discussion section.
Comment: Interpretation of Results:
The discussion tends to overstate the implications of the genetic findings, suggesting NRG2 as a susceptibility gene for ASD without sufficient supporting evidence. More detailed genetic and functional analyses are required to substantiate such claims.
Reply: Thank you for your suggestions. We have written a new paragraph to list the limitations of this study. Our findings can only be considered as prelimiary findings. Further replication studies with larger sample size, appropriate controls, and functional assay are warranted to verify our findings in the future.
In conclusion, while the study conducted is a valuable step towards understanding the genetic landscape of ASD, the concerns outlined above regarding the study's design, methodology, and interpretation of results highlight the need for more rigorous approaches.
Reply: we have included these points in our revised discussion section.

Round 2
Reviewer 2 Report
Comments and Suggestions for Authors
All comments have been addressed